# New Insights on Formyl Peptide Receptor Type 2 Involvement in Nociceptive Processes in the Spinal Cord

**DOI:** 10.3390/life12040500

**Published:** 2022-03-29

**Authors:** Mariantonella Colucci, Azzurra Stefanucci, Adriano Mollica, Anna Maria Aloisi, Francesco Maione, Stefano Pieretti

**Affiliations:** 1National Centre for Drug Research and Evaluation, Istituto Superiore di Sanità, 00161 Rome, Italy; mariantonella.colucci@iss.it; 2Department of Pharmacy, University “G. d’Annunzio” of Chieti-Pescara, 66100 Chieti, Italy; a.stefanucci@unich.it (A.S.); adriano.mollica@unich.it (A.M.); 3Department of Medicine, Surgery and Neuroscience, University of Siena, 53100 Siena, Italy; annamaria.aloisi@unisi.it; 4ImmunoPharmaLab, Department of Pharmacy, School of Medicine and Surgery, University of Naples Federico II, 80138 Naples, Italy

**Keywords:** annexin, formyl peptide receptors, nociception, pain, spinal cord

## Abstract

Formyl peptide receptor type 2 (FPR2/ALX) belongs to the formyl peptide receptors (FPRs) family clustered on chromosome 19 and encodes a family of three Class A of G protein-coupled receptors (GPCRs). A short N-terminal region, an NPXXY motif in transmembrane (TM) region 7 and an E/DRY motif that bridges TM3 and TM6 stabilizing inactive receptor conformations characterize this class of receptors. In recognizing pathogen-associated molecular patterns (PAMPs) and damage-associated molecular patterns (DAMPs), FPRs play a crucial role in innate immune responses. FPR2/ALX is highly expressed in myeloid cells, as well as in chondrocytes, fibroblasts, endothelial, epithelial and smooth muscle cells. FPR2/ALX mRNA expression was recently reported in the rat brainstem, spinal cord, thalamus/hypothalamus, cerebral neocortex, hippocampus, cerebellum and striatum. The central nervous system (CNS) distribution of FPR2/ALX suggests important functions in nociception. Thus, the present study was carried out to investigate the possible role of FPR2/ALX in nociception in mice. Intrathecal administration of the formyl peptide receptor type 1 (FPR1) agonist fMLF and the FPR2/ALX agonist BML-111 relieved nociception and these effects were reduced by contemporary administration of the FPR2/ALX antagonist WRW^4^. Furthermore, measurement of cytokines and brain-derived neurotrophic factor (BDNF) in the spinal cord of neuropathic mice demonstrated that the antinociceptive effects of BML-111 might depend on the reduction in cytokine release and BDNF in the spinal cord. These results suggest a possible role of FPR2/ALX for pain control in the spinal cord.

## 1. Introduction

N-formyl peptide receptors (FPRs) are a family of G protein-coupled receptors that play an essential role as modulators of host defense and inflammation, including cell adhesion, chemotaxis, lysosomal enzyme release and superoxide production [1]. In humans, three genes coding for FPRs, i.e., FPR1, FPR2/ALX and FPR3, have been cloned [1]. Regarding the role in inflammatory-based diseases, FPR1 promotes malignant glioblastoma progression [2] and FPR2/ALX is implicated in the pathogenesis of Alzheimer’s disease [3], wound healing, diabetes, obesity and AIDS [4]. In animals, FPRs have been found in guinea pigs, monkey, rabbits, horses, rats and mice, among others [5]. In mice, there are eight known members of the FPR gene family–*mFpr1*, *mFpr2*, *mFpr-rs1*, *mFpr-rs3*, *mFpr-rs4*, *mFpr-rs6*, *mFpr-rs7* and *mFpr-rs8*–clustered on mouse chromosome 17A3.2, while *mFpr-rs5* (*ψmFpr-rs3*) is a pseudogene that does not code for a functional receptor [1].

The FPR family is “promiscuous” by reason of the structural diversity of the ligands, ranging from eicosanoids, lipopeptides, peptides and synthetic non-peptide compounds. N-formylated peptides are the most commonly studied ligands of this receptor family, whose name derives from them. The shortest formyl peptide with full agonistic FPR1 activities in humans is the *E. coli*-derived formyl-MLF (fMLF), a potent chemoattractant that recruits and guides leukocytes to the site of bacterial infection and damaged tissues. Human FPR2/ALX is a low affinity receptor for fMLF, since it binds with relatively high affinity and responds better to N-formylated peptides, such as fMIFL and fMIVIL, or those carrying positive charges at the C-terminus, such as fMLFK and fMLFIK [6]. Other non-formylated peptides or proteins of microbial, endogenous or synthetic origin have been shown to be agonists for FPR1 and/or FPR2/ALX [1]. For example, Annexin1 (ANXA1) belongs to the family of Ca^2+^ and phospholipid-binding proteins named annexins [7]. endogenous ANXA1 mediates the anti-inflammatory effects of glucocorticoids. The most important of these are the inhibition of phospholipase A2 activity and neutrophil migration. ANXA1 mediates its pharmacological effects by binding to FPR2/ALX, and also the ANXA1 N-terminal-derived peptides Ac2-26, Ac2-12 and Ac2-6 induced FPR1 and/or FPR2/ALX activation in both humans [7] and mice [8]. 

The role of FPRs in a host–defense setting has been extensively studied and well defined, and recent research has highlighted the possible role of FPRs in the nervous system [9]. However, few data have been reported on the possible participation of FPRs in nociception. In the first study in this field, Ferreira and collaborators demonstrated that Ac2-26 inhibits the hyperalgesic effects induced by TNF-α and PGE2 administration and the release of these two mediators in a murine macrophage-like cell-line stimulated with LPS [10]. In a later study [11], we demonstrated the selective anti-nociceptive effects of Ac2-26 and Ac2-12 in the formalin test but not in the hot plate or tail flick tests. More importantly, after peripheral and central administration, the FPR1 agonist fMLF displayed the same effects observed after Ac2-26 administration, and the FPRs antagonist Boc-1 blocked Ac2-26, Ac2-12 and fMLF effects in this assay of inflammatory nociception [11]. Further studies demonstrated that ANXA1 null mice were more susceptible to nociception induced by intraperitoneal injection of acetic acid than wild-type, suggesting that ANXA1 might endogenously modulate nociception [12]. In addition, increased levels of PGE2 in the spinal cord of ANXA1 null mice compared with wild-type suggested that ANXA1 modulates nociceptive processing at the spinal cord level by downregulating PGE2 spinal nociceptive facilitation [12]. Therefore, these results suggest that endogenous ANXA1 and FPRs are not involved in thermal nociception, whereas might have a role in inflammatory pain induced by chemical stimuli such as formalin or acetic acid. 

Other mechanisms might be involved in the antinociceptive effects mediated by FPRs. For example, it has been reported that inflammatory pain evoked by intraplantar injection of complete Freund’s adjuvant (CFA) is reduced by local opioid peptide release triggered by fMLF-FPRs in rat peripheral neutrophils [13]. More recently, intrathecal Ac2-26 or BML-111–an FPR2/ALX agonist–decreased hyperalgesia in CFA-induced inflammatory pain [14]. Both ANXA1 and FPR2/ALX are expressed in dorsal root ganglion (DRG) and spinal dorsal horn, neuronal areas linked to the development of nociception [14,15]. 

These data suggest an involvement of FPRs in the modulation or control of nociception in the spinal cord. Nonetheless, to our knowledge, no studies have investigated the intrathecal administration effects of FPRs agonists in nociception induced by different stimuli. In the present study, we used thermal nociception (hot plate and tail flick test), short- and long-term inflammatory nociception (formalin test and carrageenan-induced hyperalgesia), and neuropathic nociception (chronic constriction of the sciatic nerve) to further investigate the possible involvement of FPR1 and FPR2/ALX in spinal cord nociception after intrathecal administration of selective agonists in mice. Research from several groups has revealed an important role of both pro- and anti-inflammatory cytokines and brain-derived neurotrophic factor (BDNF) in neuropathic and other chronic pain states [16]. In the context of neuropathic pain, TNF-α is perhaps the most widely studied pro-inflammatory cytokine, but also other molecules as IL-1β, IL-6, and IL-17 are garnering increased interest. Thus, in our model of neuropathic pain, TNF-α, IL-1β, IL-6 and BDNF release from the spinal cord was investigated to ascertain, for the first time, whether FPR agonists might affect nociception by modifying the levels of inflammatory cytokines and BDNF in the spinal cord.

## 2. Materials and Methods

### 2.1. Drugs and Treatment Procedure

fMLF (MW, 437.55), BML-111 (MW, 192.2) Boc-1 (MW, 509.66) and WRW^4^ (MW, 1104.28) were purchased from Tocris-BioTechne SRL (Milan, Italy) and the other materials were obtained from Sigma–Aldrich SRL (Milan, Italy). In order to perform the intrathecal (i.t.) treatment, aliquots of the peptide stock solutions in DMSO were used for dilution in saline (DMSO:saline 1:3, *v*/*v*) and administered immediately after sonication. The i.t. injection was made into the L5-L6 intervertebral space of mice anesthetized with isoflurane (5%) using a 25-μL micro-syringe connected to a 27-gauge stainless steel needle as previously described by Hylden and Wilcox [17]. A flick of the tail was considered indicative the needle had entered the subarachnoid space. Drug solutions were injected over a period of more than 10 s. The needle was removed after a further 10 s period, in order to ensure solution retention. The injection volume was 5 μL/mouse. In all experiments, the drugs were administered i.t. at the doses of 0.2 or 2 nmol.

### 2.2. Animals and Experimental Procedures

CD-1 male mice (Harlan, Italy) of 3–4 weeks (25–30 g) were used for all the experiments. Before the experimental sessions, mice were housed in colony cages under standard light, temperature and relative humidity conditions for at least 1 week. The experimental protocols performed in the present study were in accordance with Italian Legislative Decree 27/92 and approved by the local ethics committee (approval number: 277/2012-B). 

### 2.3. Hot Plate Test

The hot plate (25 × 25 cm metal plate, Socrel Mod. DS-37, Ugo Basile, Varese, Italy) was set to a temperature of 55 ± 0.1 °C on which a plastic cylinder 20 cm diameter, 18 cm high, was placed. The time of latency (s) was recorded from the moment the animal was inserted into the cylinder until it first licked its paws or jerked them off the hot plate, or jumped off the hot plate or the latency exceeded the cut-off time of 30 s. The baseline hot plate latency was recorded at 90, 60 and 30 min before treatment and at 15, 30, 45 and 60 min after treatment [11]. Baseline latency was 5–13 s.

### 2.4. Tail Flick Test

The tail flick unit (Socrel Mod DS-20, Ugo Basile, Italy) consisted of an infrared radiant light source (100 W bulb, 20 V) focused onto a photocell utilizing an aluminum parabolic mirror. A glove was used for gently hand-restrain the mouse during the trials. Radiant heat was focused on the middle part of the tail and the latency time (s) the mouse took to flick its tail was recorded. A cut-off time of 15 s was imposed. Baseline tail flick latency was recorded at 90, 60 and 30 min before treatment and at 15, 30, 45 and 60 min after treatment [11]. Baseline latency was 3–6 s.

### 2.5. Formalin Test

The procedure has been described previously [18]. Subcutaneous (s.c.) injection of formalin (1% in saline, 20 μL/paw) into the mouse’s hind paw induces nociceptive behavioral responses, such as licking or biting the injected paw, which are considered indices of pain. The nociceptive response consists of an early phase occurring from 0 to 10 min after formalin injection due to stimulation of peripheral nociceptors, followed by a prolonged late phase occurring from 10 to 40 min that reflects the response to inflammatory pain. The day of the test, the mouse was placed in a Plexiglas cage (30 × 14 × 12 cm) 1 h before the formalin administration to allow it to acclimatize to its surroundings. Immediately after formalin injection, the mouse was returned to the cage and nociceptive behavior was continuously measured using a stopwatch for each 5 min block for 40 min. The total time (s) the animal spent licking or biting its paw during the formalin-induced early and late phase of nociception was recorded. 

### 2.6. Carrageenan-Induced Thermal Hyperalgesia

The plantar test (Ugo Basile, Italy) was used to measure the sensitivity to a noxious heat stimulus after carrageenan administration [18]. A radiant heat source was directed on the mouse’s footpad until its withdrawal, foot drumming or licking. The withdrawal time latency was automatically recorded upon the removal of the hind paw. Animals were acclimatized to their environment for 1 h before the measurements of paw withdrawal latency (PWL) and the heat intensity was adjusted to obtain a baseline between 10 and 15 s. The cut off time of this test was set at 30 s to prevent tissue injury in the mice. Three readings were obtained from each paw and were averaged. Animals were first tested to determine their baseline PWL; 2 h later, each animal received an i.pl. injection of 50 μL of 1% carrageenan into the right hind paw. After the carrageenan injection, the PWL(s) of each animal was determined again at 1, 2, 4, 6 and 24 h. 

### 2.7. Neuropathic Thermal Hyperalgesia

The chronic constriction injury (CCI) model, based on a previous description [19], was adopted with some modifications [20]. The surgery was performed in mice after they were deeply anesthetised with chloral hydrate–xylazine (400 + 10 mg, 10 mL/kg i.p.). The right sciatic nerve was exposed and proximal to the nerve trifurcation, it was then loosely ligated with four ligatures with 9-0 non-absorbable black nylon monofil (S&T, Neuhausen am Rheinfall, Switzerland). The ligations were approximately 1 mm apart. After surgery, mice were allowed to recover for 3 days. Thermal hyperalgesia was then measured by subjecting the CCI mice to the plantar test, as described in the previous section. Sham-operated animals (sciatic nerve exposure without ligation) were used as controls.

### 2.8. Cytokines and BDNF Assays

CCI mice were decapitated ten days after surgery [20] and the spinal cord was collected and stored in liquid nitrogen. Then, the tissues were homogenized in 100 mg/mL of PBS solution (0.4 M NaCl, 0.05% Tween 20, 0.5% BSA, 0.1 mM phenylmethylsulfonyl fluoride, 0.1 mM benzethonium chloride, 10 mM EDTA and 20 KIU aprotinin). The homogenates were centrifuged at 10,000× *g* for 10 min at 4 °C, and the supernatants were stored at −20 °C. IL1β, IL-6, TNF-α and BDNF concentrations were determined at a 1:3 dilution in PBS containing 0.1% BSA, using an ELISA kit according to the manufacturers’ instructions (R&D Systems, Minneapolis, MN, USA and Pharmingen, San Diego, CA, USA). The concentration of each cytokine and BDNF was calculated using a standard curve and expressed in pg/mL.

### 2.9. Data Analysis and Statistics

Data were analyzed using two-way ANOVA followed by Dunnett’s multiple comparisons test. Formalin test data were analyzed using one-way ANOVA followed by Dunnett’s multiple comparisons test. GraphPad Prism 6.0 software (San Diego, CA, USA) was used for the analyses. Differences between means were considered statistically significant at *p* ≤ 0.05. Sample size was chosen to ensure alpha 0.05 and power 0.8. Animal weight was used for randomization and group allocation to reduce unwanted sources of variations by data normalization. In vivo and in vitro studies were carried out to generate groups of equal size, using randomization and blinded analysis. No responsive mice were excluded from the analysis. For each experiment, data and the statistical test are specified in the figure legends.

## 3. Results

### 3.1. Effects of FPR Agonists and Antagonists in the Hot Plate and Tail Flick Tests

In the first series of experiments, the effects of the FPR agonists fMLF and BML-111 were investigated in animal models of acute nociception induced by thermal stimuli in the hot plate and tail flick tests. Drugs were injected i.t. 30 min before the test started. fMLF administered at the doses of 0.2 and 2 nmol did not change the response of mice in either the hot plate or tail flick test (Figure 1A,B). Likewise, BML-111 administered at the doses of 0.2 and 2 nmol did not change the response to thermal nociception in either test (Figure 1C,D).

After the above experiments, we determined whether the FPR antagonists Boc-1 and WRW^4^ would change the nociceptive response to thermal stimuli in the mice. At the dose of 0.2 nmol, neither Boc-1 nor WRW^4^ was able to change the response to thermal stimuli in the hot plate (F_(2, 18)_ = 0.007256; *p* = 0.9928, n.s.) and tail flick test (F_(2, 18)_ = 0.1648; *p* = 0.8493, n.s.) (data not shown).

### 3.2. Effects of FPR Agonists and Antagonists in the Formalin Test

Drugs were injected i.t. 30 min before the formalin test started. In the first series of experiments, the effects of fMLF at the doses of 0.2 and 2 nmol were investigated. At the lower dose, fMLF did not reduce formalin-induced licking in either the early or late phase of the test (Figure 2A). At the higher dose, fMLF was able to significantly reduce the licking, both in the early and late phase (Figure 2A).

The effects of BML-111 were then investigated. Contrary to what was observed after administration of fMLF, BML-111 reduced licking in both phases of the test even at the 0.2 nmol dose (Figure 2B). At 2 nmol, BML-111 strongly reduced the licking behavior in both the early and late phase (Figure 2B). 

The effects of FPR antagonists Boc-1 and WRW^4^ at the dose of 0.2 nmol were then investigated. Boc-1 and WRW^4^ did not change the formalin-induced effects in either the early (F_(2, 18)_ = 1.777; *p* = 0.1975, n.s.) or late phase (F_(2, 18)_ = 1.833; *p* = 0.1886, n.s.) of the test (data not shown). We then investigated whether the antinociceptive effects of fMLF or BML-111 might be affected by contemporary administration of Boc-1 or WRW^4^. Boc-1 (0.2 nmol) administered with fMLF (2 nmol) did not change the antinociceptive effects of fMLF in the early and late phase of the test (Figure 2C). The same lack of effect in both phases was observed when Boc-1 was administered with BML-111 (2 nmol) (Figure 2D). When the FPR2/ALX antagonist WRW^4^ (0.2 nmol) was administered with fMLF (2 nmol), it reduced the fMLF-induced antinociceptive effects in both phases (Figure 2C). The antinociceptive effects induced by BML-111 were also strongly antagonized by WRW^4^, both in the early and late phase (Figure 2D).

### 3.3. Effects of FPR Agonists and Antagonists on Carrageenan-Induced Thermal Hyperalgesia

The effects of fMLF and BML-111 on the nociceptive threshold were further investigated in the carrageenan-induced inflammation pain model. The development of thermal hyperalgesia, measured as a reduction in PWL to thermal stimuli, begins 2 h after carrageenan injection and reaches its peak 4–6 h later (Figure 3). In this test, drugs were injected i.t. 30 min before the PWL measurements, performed 6 h after carrageenan injection. Based on the results of the previous experiments, we decided to minimize the number of animals, treating them with fMLF or BML-111 only at the dose of 2 nmol. 

fMLF did not increase the carrageenan-induced nociceptive threshold reduction (Figure 3A), whereas BML-111 increased the nociceptive threshold at both 6 and 24 h after carrageenan administration (Figure 3A). The i.t. administration of Boc-1 or WRW^4^ did not change the effects of carrageenan 6 h after its administration. (F_(2, 18)_ = 0.8210; *p* = 0.4558, n.s.) (data not shown). Since BML-111 reduced the effects of carrageenan on the nociceptive threshold, we investigated whether this effect could be antagonized by Boc-1 or WRW^4^. The contemporary administration of Boc-1 (0.2 nmol) with BML-111 (2 nmol) did not change the BML-111 effects, whereas the WRW^4^ antagonist (0.2 nmol) was able to reduce the BML-111 antinociceptive effects at 6 and 24 h after carrageenan administration (Figure 3B).

### 3.4. Effects of FPR Agonists and Antagonists on Neuropathic Hyperalgesia

A model of chronic neuropathic pain was established in mice by sciatic nerve ligation and the effect of fMLF and BML-111 on hyperalgesia was evaluated in terms of paw withdrawal latency from thermal nociceptive stimulus (PWL). A decrease of paw withdrawal threshold induced by sciatic nerve ligation, was observed in the ipsilateral but not the contralateral paw. On the seventh day post-surgery, the nociceptive threshold reached its lowest point and maintained at this value for the rest of the testing period, indicating that the neuropathic pain model was successfully induced. fMLF or BML were administered i.t. at a dose of 2 nmol 30 min before the PWL measurements, performed 10 days after the surgery. fMLF and BML-111 increased the time to respond to thermal stimuli in the neuropathic animals. fMLF induced a slight, non-significant increase in PWL, whereas BML-111 strongly increased PWL (Figure 4A). The i.t. administration of Boc-1 or WRW^4^ did not change the effects induced by sciatic nerve damage 10 days after the surgery (F_(2, 18)_ = 1.012; *p* = 0.3833, n.s.) (data not shown). 

We then investigated the effects of Boc-1 and WRW^4^ on BML-111-induced antinociception (Figure 4B). The contemporary administration of Boc-1 (0.2 nmol) together with BML-111 was not able to change the antinociceptive effects induced by the FPR2/ALX agonist, whereas WRW^4^ (0.2 nmol) reduced the antinociceptive effects induced by BML-111 (Figure 4B).

### 3.5. Measurement of Cytokines and BDNF in the Spinal Cord

The effects of fMLF and BML i.t. treatment at the dose of 2 nmol on cytokine and BDNF release from the spinal cord of neuropathic animals were investigated. In neuropathic animals, an increase in cytokine and BDNF release occurs in the spinal cord [16], and the same effects were observed in our animals after sciatic nerve ligation (Figure 5A–D). When TNF-α release was measured in neuropathic animals after fMLF treatment, the peptide induced a slight but non-significant reduction in TNF-α levels (Figure 5A). BML-111 instead induced a significant reduction in TNF-α levels (Figure 5A). Neither fMLF nor BML-111 were able to change the neuropathy-induced effects on IL-β levels in the spinal cord (Figure 5B). IL-6 levels were reduced after i.t. treatments with fMLF and BML-111, although only BML-111 significantly reduced the IL-6 increase induced after sciatic nerve ligation (Figure 5C). BDNF levels increased in neuropathic animals, and in animals treated with fMLF, the BDNF release was unchanged after peptide treatment (Figure 5D). In animals treated with BML-111, the BDNF increase induced by neuropathy was reduced (Figure 5D).

## 4. Discussion

In animal models examining thermal-induced nociception (hot plate and tail-flick tests), the measured parameter is latency of the nocifensive reaction evoked by suprathreshold heat intensity. For analgesic substances, the nocifensive behavior induced by thermal stimuli is affected principally by centrally acting drugs such as opioids [21]. In agreement with our previous observations [11], the agonists fMLF and BML-111 and the antagonists Boc-1 and WRW^4^ were not able to modify the behavioral response induced by thermal nociceptive stimuli. 

Nociception can also be induced by chemical agents in order to preclinically evaluate potential analgesic drugs, and the formalin test is one of the most commonly used procedures [21]. Both fMLF–at a relatively high dose–and BML-111–at low and high doses– reduced the licking response in the early and late phase of the formalin test and this effect was reduced by contemporary administration of the FPR2/ALX antagonist WRW^4^. The formalin test can be considered a short-term inflammatory pain model. Studies on inflammatory pain use compounds with strong antigenic potential, such as carrageenans, which are sulfated polysaccharides extracted from seaweed. After intraplantar injection, carrageenans activate and sensitize the nociceptive system, inducing both thermal and mechanical allodynia and hyperalgesia for at least several hours. In the present study, i.t. treatment with BML-111 reversed the carrageenan-induced hyperalgesia, whereas fMLF was ineffective. Boc-1 did not change the nociceptive effects of BML-111, whereas WRW^4^ effectively antagonized its effects.

Neuropathic pain can be a consequence of peripheral nerve injury, diabetes, infectious diseases and exposure to neurotoxic compounds, or can be of central origin. Most murine models of peripheral nerve injury target the sciatic nerve. The most commonly used model is chronic constriction injury (CCI), which induces mechanical allodynia and changes in sensitivity or response to thermal stimuli in the ipsilateral paw. To our knowledge, no data are available on the effects of FPR agonists in neuropathic pain models. In our experiments, BML-111 strongly counteracted the CCI-induced thermal hyperalgesia, while WRW^4^ was able to reduce the BML-111-induced antinociceptive effect. Neuropathic pain induces plastic changes at the site of nerve injury, the dorsal root ganglia (DRGs) and the dorsal horn of the spinal cord. The endothelial damage and increased neuronal activity result in peripheral recruitment of monocytes/macrophages and spinal cord activation of microglia, which release mediators such as cytokines and BDNF. This leads to sensitization of neurons, enabling positive feedback that sustains chronic pain [16]. In the present experiments and as already reported [20], CCI increased TNF-α, IL-1β, IL-6 and BDNF release from the spinal cord and BML-111 effectively counteracted this effect, suggesting a possible mechanism of action for the antinociceptive effects of fMLF and BML-111. 

In a previous investigation on the possible involvement of ANXA1 in the inhibition of nociception via FPR2/ALX in the DRGs in a model of inflammatory pain, the subcutaneous injection of complete Freund’s adjuvant (CFA) into the rat hind paw resulted in upregulation of ANXA1 in the L4/5 DRGs [14]. Thermal hyperalgesia and mechanical allodynia induced by CFA were attenuated after i.t. administration of Anxa12-26 and BML-111, and these treatments upregulated ANXA1 expression in the L4/5 DRGs. These last effects were suppressed when FPR2/ALX was inhibited by i.t. Boc-1 [14]. Similar results were obtained in our experiments, even if Boc-1 did not antagonize the effects induced by fMLF and BML-111. However, Pei et al. [14] used much higher doses of Boc-1 (10 and 100 μg) than those used in our experiments (0.2 nmol, about 100 ng), and at high doses Boc-1 can also interact with FPR2/ALX. We decided instead to use a very low dose of Boc-1 in an attempt to block only FPR1. Thus, all these results suggest a system at the spinal cord level that can play a role in controlling nociception through the endogenous action of ANXA1 and the FPR2/ALX receptors. Other recent data [22] seem to confirm this hypothesis. In a model of opioid-induced hyperalgesia, mechanical allodynia and thermal hyperalgesia was observed after remifentanil administration, followed by an increase in spinal ANXA1 and CXCL12/CXCR4 expression. Ac2-26 injected i.t. reduced the effects of remifentanil, facilitated ANXA1 production and inhibited upregulation of CXCL12/CXCR4 levels and NR2B-containing N-methyl-d-aspartate receptor (NMDAR) phosphorylation. Moreover, pretreatment with the selective CXCR4 antagonist AMD3100 reduced hyperalgesia and NR2B-containing NMDAR phosphorylation [22]. This further suggests a role for ANXA1 in nociception control in the spinal cord and indicates the participation of spinal CXCL12/CXCR4 and NR2B-containing NMDAR pathways in the ANXA1-induced anti-hyperalgesic effects, at least in opioid-induced hyperalgesia. 

Other speculative hypotheses on the role of ANXA1 and FPR2/ALX in the control of nociception at the spinal cord level can be formulated. A recent study [23] reported that ANXA^−/−^ mice exhibited significant sensitivity to noxious heat, capsaicin, formalin and CFA. In ANXA1^−/−^ cultured DRG neurons, an increase in capsaicin-induced Ca^2+^ response, TRPV1 currents and neuronal firing was recorded. Furthermore, Ac2-26 robustly increased intracellular Ca^2+^, inhibited the TRPV1 current, activated PLCβ and promoted CaM-TRPV1 interactions that were attenuated by the FPR2/ALX antagonist Boc-2 [23]. All these effects provide new evidence on the possible role for ANXA1-FPR2/ALX signaling in the spinal cord driving, via TRPV1, the control of nociception. However, Ac2-26 has an effect on the anti-inflammatory response mediated by astrocytes, and microglia and astrocytes of the central nervous system play a fundamental role in the development and maintenance of chronic pain. Ac2-26 inhibited astrocyte migration, reduced the production of TNF-α, IL-1β, monocyte chemoattractant protein-1 (MCP-1) and macrophage inflammatory protein-1 (MIP-1α), and upregulated GSH reductase mRNA and GSH levels in LPS-induced astrocytes in vitro. The involvement of the p38 and JNK-MAPK signaling pathway in this process was demonstrated, but its appeared not dependent on the NF-κB pathway [24]. Furthermore, p38 and JNK inhibitors mimicked the effects of Ac2-26, whereas a p38 and JNK activator anisomycin partially reversed its function. In vivo, Ac2-26 induced antinociceptive effects after i.t. injection in CFA-induced inflammatory pain, prevented CFA-induced GFAP-IF upregulation and decreased the CFA-induced increase in TNF-α, IL-1β, MCP-1 and MIP-α mRNA expression as well as elevated GSH levels in the spinal cord [24]. These results, together with those from our experiments on cytokine releases in the CCI model, suggest that the antinociceptive effects of FPR2/ALX agonists at the spinal cord level may also depend on the reduction in cytokine release from astrocytes activated by prolonged nociceptive stimuli. Finally, the ANXA1-FPR2/ALX network might also act at the start of the nociceptive stimuli, since it has been reported in an animal model of persistent peripheral inflammation that subcutaneous BML-111 injection for five consecutive days reduced mechanical hyperalgesia. These effects also appear to be mediated by FPR2/ALX since WRW^4^ prevented the anti-hyperalgesic effect induced by BML-111 [25]. 

Regardless of what the mechanism may be, our data confirm the antinociceptive effects of FPR2/ALX agonists at the spinal cord level in inflammatory nociception and now, observed for the first time, in a neuropathic pain model, indicating their translational potential. These effects could depend on the release of cytokines and BDNF from the spinal cord induced by FPR2/ALX agonists.

## Figures and Tables

**Figure 1 life-12-00500-f001:**
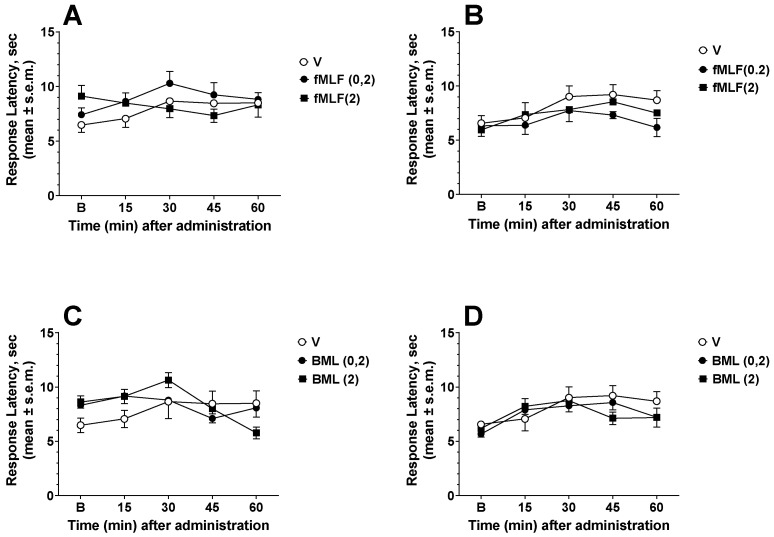
fMLF and BML-111 (BML) effects in the hot plate (**A**,**C**) and tail flick (**B**,**D**) tests. fMLF and BML were administered i.t. at the doses of 0.2 and 2 nmol. fMLF and BML did not change the mice’s responses to thermal stimuli. Data were analyzed using two-way ANOVA followed by Dunnett’s multiple comparisons test. N = 7.

**Figure 2 life-12-00500-f002:**
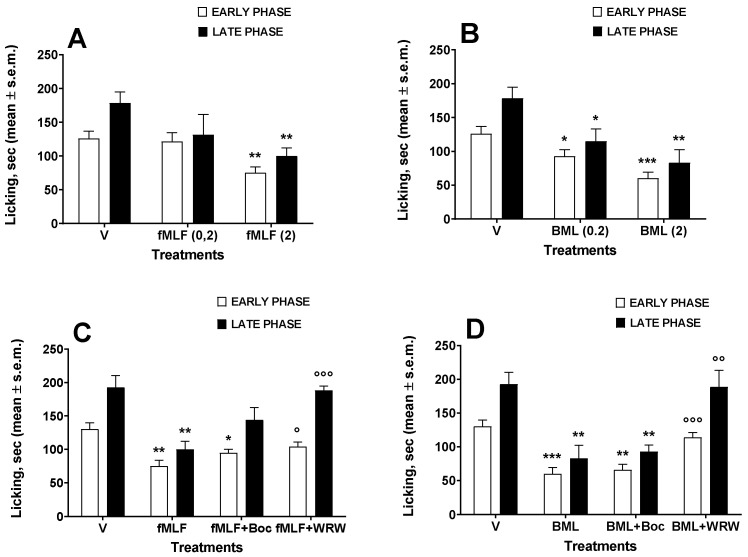
fMLF (**A**) and BML-111 (BML) (**B**) effects in the formalin test. fMLF and BML-111 were administered i.t. at the doses of 0.2 and 2 nmol. In the antagonism experiments with Boc-1 (Boc) and WRW^4^ (WRW), the antagonists were administered at a dose of 0.2 nmol contemporarily with fMLF (**C**) or BML (**D**) administered at a dose of 2 nmol. * is for *p* < 0.05, ** is for *p* < 0.01 and *** is for *p* < 0.001 vs. V (vehicle-treated animals); ° is for *p* < 0.05, °° is for *p* < 0.01, °°° is for *p* < 0.001 vs. fMLF- or BML-treated animals. Data were analyzed by one-way ANOVA followed by Dunnett’s multiple comparisons test. N = 7.

**Figure 3 life-12-00500-f003:**
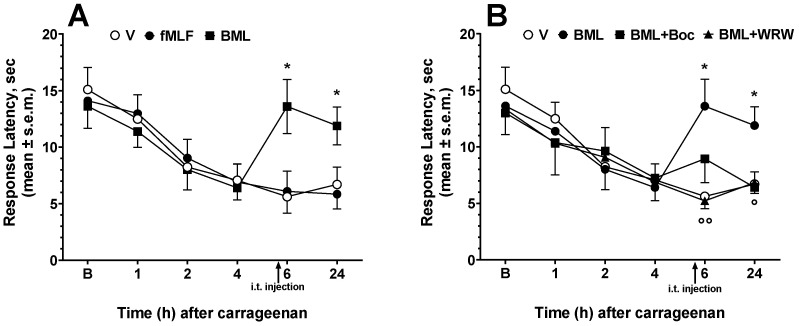
fMLF and BML-111 (BML) effects on carrageenan-induced thermal hyperalgesia. fMLF and BML-111 were administered i.t. at a dose of 2 nmol (**A**). In the antagonism experiments with Boc-1 (Boc) and WRW^4^ (WRW), the antagonists were administered at a dose of 0.2 nmol contemporarily with BML (**B**). * is for *p* < 0.05 vs. V (vehicle-treated animals); ° is for *p* < 0.05, °° is for *p* < 0.01 vs. BML-treated animals. Data were analyzed by two-way ANOVA followed by Dunnett’s multiple comparisons test. N = 7.

**Figure 4 life-12-00500-f004:**
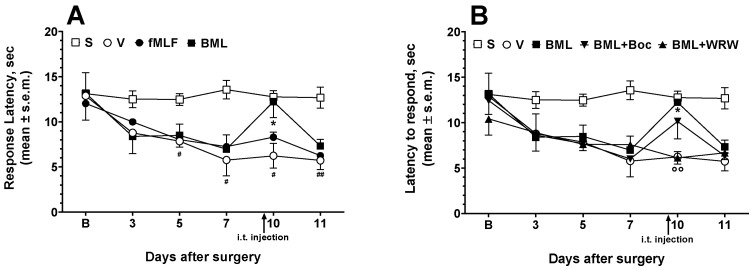
fMLF and BML-111 (BML) effects on neuropathic thermal hyperalgesia. fMLF and BML-111 were administered i.t. at a dose of 2 nmol (**A**). In the antagonism experiments with Boc-1 (Boc) and WRW^4^ (WRW), the antagonists were administered at a dose of 0.2 nmol contemporarily with BML (**B**). * is for *p* < 0.05 vs. V (vehicle-treated animals); °° is for *p* < 0.01 vs. BML-treated animals; ^#^ is for *p* < 0.05 and ^##^ is for *p* < 0.01 vs. S (sham-operated animals). Data were analyzed by two-way ANOVA followed by Dunnett’s multiple comparisons test. N = 7.

**Figure 5 life-12-00500-f005:**
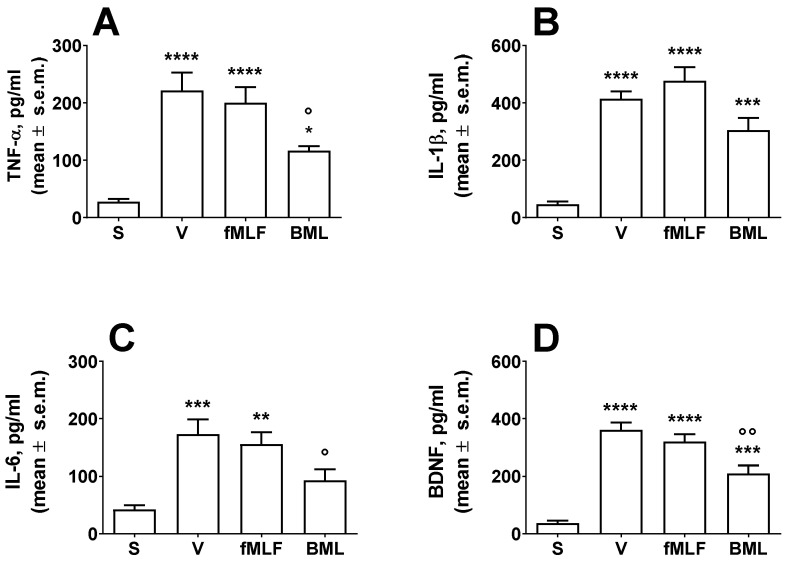
Effects of fMLF and BML-111 (BML) on TNF-α (**A**), IL-1β (**B**), IL-6 (**C**) and BDNF (**D**) levels in the spinal cord in neuropathic animals. fMLF and BML-111 were administered i.t. at a dose of 2 nmol. ** is for *p* < 0.01, *** is for *p* < 0.001 and **** is for *p* < 0.0001 vs. S (sham-operated animals); ° is for *p* < 0.05 and °° is for *p* < 0.01 vs. V (vehicle-treated animals). Data were analyzed by one-way ANOVA followed by Dunnett’s multiple comparisons test. N = 6.

## Data Availability

Data presented in this paper are available upon request to the corresponding authors.

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
