# Peer review of "New Insights on Formyl Peptide Receptor Type 2 Involvement in Nociceptive Processes in the Spinal Cord"

_life, 2022, doi:10.3390/life12040500_

Round 1

Reviewer 1 Report

The authors explored the role of formyl peptide receptor type 2 (FRP2) in nociception. The data suggest that FRP2 activation suppresses pain response, at least in some experimental models. The data also suggest that agonist BML-111 is more potent in vivo than fMLF, and that antagonist WRW is more potent than Boc-1. The discussion of possible molecular mechanisms is largely speculative, and that should be emphasized.

Overall, this is an interesting study, which can be further improved by inclusion of untreated controls in all experiments to allow the reader judge whether antagonists blocked the action of agonists completely or partially. Additional changes would also improve the manuscript

  1. Data on untreated controls should be included in Figs. 2 and 3.
  2. Remove “Fig X reports…” throughout (Fig. X in parenthesis would suffice.
  3. There is no need to recapitulate all results in the discussion. The discussion can be shortened by 30-40%.
  4. The intro can be shortened by 10-15%.
  5. Some editing, preferably by a native speaker, is needed: line 65, “are initiated with” should be “start with”; line 83, “to be agonists” should be “as agonists”; line 90, “small molecule” should be “small molecules”; ibid., delete “highly”; line 92. “commercially stable” should be “commercially available”; ibid., “in confront to” should be “than”; etc. (too many to point them all out).

Author Response

We thank the reviewer for the helpful comments and suggestions. Our responses to suggestions and comments are shown below in italics. Based on the reviewers' comments and suggestions, we have made significant changes to the text and figures that greatly improved our manuscript.

The authors explored the role of formyl peptide receptor type 2 (FRP2) in nociception. The data suggest that FRP2 activation suppresses pain response, at least in some experimental models. The data also suggest that agonist BML-111 is more potent in vivo than fMLF, and that antagonist WRW is more potent than Boc-1. The discussion of possible molecular mechanisms is largely speculative, and that should be emphasized.

Following the reviewer's suggestions, we changed the discussion as suggested, emphasizing its speculative character regarding the mechanism underlying the effects of the substances used in our experiments. Please, see the discussion section in the revised form of the manuscript.

Overall, this is an interesting study, which can be further improved by inclusion of untreated controls in all experiments to allow the reader judge whether antagonists blocked the action of agonists completely or partially. Additional changes would also improve the manuscript. Data on untreated controls should be included in Figs. 2 and 3.

We thank the reviewer for his comment, which makes us understand that we were not clear in exposing our data. In the revised version of the manuscript, we added the results obtained from the vehicle-treated animals in all panels of the figures. If by “untreated control” you mean “naive animals”, we have not included these data, since nociceptive threshold did not change in animals treated with the vehicle in confront to naive animals. Below we report the results of the statistical analyses performed by comparing the naive animal groups vs the animals treated with the vehicle. Hot plate test: F(1, 12)=0,7094, P=0,4161; Tail flick test: F(1, 12)=0,01850, P=0,8941; Formalin test, early phase: t12=1,748, P=0,1059; Formalin test, late phase: t12=1,154, P=0,2709; Carrageenan-induced thermal hyperalgesia: F(1, 12) = 0,8227, P=0,3822; Neuropathic thermal hyperalgesia: in this case, the comparison with sham-untreated animals is reported in the new version of Figure 4; Measurement of cytokines and BDNF: In this case, the comparison with sham-untreated animals is reported in the new version of Figure 5.

Remove “Fig X reports…” throughout (Fig. X in parenthesis would suffice.

We have removed the phrase “ Fig X reports”, as you have suggested throughout the text of the new version of the manuscript.

  1. There is no need to recapitulate all results in the discussion. The discussion can be shortened by 30-40%.

We have reduced the discussion by 40%, as you have suggested. Please, see the discussion section in the new version of the manuscript

  1. The intro can be shortened by 10-15%.

We have reduced the introduction by 10-15%, as you have suggested. Please, see the introduction section in the new version of the manuscript

  1. Some editing, preferably by a native speaker, is needed: line 65, “are initiated with” should be “start with”; line 83, “to be agonists” should be “as agonists”; line 90, “small molecule” should be “small molecules”; ibid., delete “highly”; line 92. “commercially stable” should be “commercially available”; ibid., “in confront to” should be “than”; etc. (too many to point them all out).

As you suggested, the new version of the manuscript was revised by an expert native English.

Reviewer 2 Report

see word file

Author Response

We thank the reviewer for his review and helpful comments and suggestions. Our responses to suggestions and comments are shown below in italics. Based on the reviewers' comments and suggestions, we have made significant changes to the text and figures that greatly improved our manuscript.

Colucci et al. report on the administration of N-formyl peptide receptor (FPR) agonists and antagonists and their effect on nociception in mice. While I believe, that there is quite some interesting information it is rather complicated to extract the significance of this work. The structure of the introduction should be reshaped to really point out the relevance of this topic, the known facts that actually concern the experiments that were undertaken and the open questions that lead to the proposed study. In detail I would like to see the following changes:

  • Please include what is known for FPR and nociception (besides your own study) and what remains to be identified? What is the significant contribution of this study to the field?

As you suggested, in the Introduction section of the new version of the manuscript, we included what is known on FPR and nociception besides our first study, what remains to be identified and what is the significant contribution of this study in the field. Please, see page 2 - lines 62-93, in the revised version of the manuscript.

  • Please mention the purpose of the different pain models used and how they help answering the open questions identified in the previous section?

As you suggest, we mentioned the reasons for using different pain models and how these can help us clarify the role of FPRs in pain. Please, see pages 2-3 – lines 94-109, in the revised version of the manuscript.

  • There is an overly extensive description of FPR orthologues, which is not relevant for the story. Please shorten.

As you suggest, we have reduced the description of the FPR orthologs, in the revised version of the manuscript. Please, see pages 1-2 – lines 43-47, in the revised version of the manuscript.

  • Instead of focussing on the characteristics of the known agonists, please mention which ones have been used to study nociception in previous studies and explain your choice of agonists and antagonists for this study. The characteristics of all known agonists/antagonists can be used in the discussion to point out differences in their nociceptive profile.

As you suggest, in the new version of the manuscript, we reviewed the discussion discussing the effects of the agonists and antagonists used in previous studies and explained the choice of those used in our experiments and the observed differences between our study and previous ones. Please, see pages 10-11 – lines 368-420, in the revised version of the manuscript.

The title is confusing: FPR2 is not a subfamily and it sounds more like a review article. The title should be more representative of what the study provides; the pharmacological targeting of FPR2 to study its role in nociception.

As you suggest, in the revised version of the manuscript we have changed the title in “New insights on formyl peptide receptor type 2 involvement in nociceptive processes in the spinal cord” to make it more relevant to the purpose of our study and its findings.

Experimental issues:

  • Please explain what they choice of applied doses of agonists and antagonists was based on. Especially the applied dose of Boc-1 appears to be insufficient. The authors point out that at higher concentrations Boc-1 loses his FPR1-specific antagonistic effects but I do not see an assay that shows significant effects of this antagonist at least as proof of principle.

In our study, we wanted to use a dose of antagonist with no effect per se, in order to obtain only an antagonism towards the effects induced by the agonists. At higher doses these antagonists can produce effects on their own (Pei et al., Br J Anaesth 2011 Dec;107:948-958), which could confuse the results obtained in the co-administration tests. However, Boc-1 was able to antagonize the effects of fMLF and BML-111 in the formalin test.

  • Statistics: please use accepted posthoc tests following the ANOVA. T-test is not one of them.

As you suggest, we repeated the statistical analyzes using Dunnett's multiple comparisons test.

  • All data that is mentioned should be shown at least in a supplemental file.

In the revised version of the manuscript, we have eliminated all redundant data (such as the results of statistical analyzes) and we do not consider it appropriate to propose the deleted data in an additional section.

  • Please provide data to prove the successful induction of neuropathic pain described in section 3.4 (p.8, ll. 309-312)

As you suggested, the comparison with sham-operated animals is reported in the new version of Figure 4 that demonstrates the reduction of nociceptive threshold in CCI animals.

  • Please provide data for the non-affected control in the cytokine measurements (Fig.5).

As you suggested, the comparison with sham-operated animals is reported in the new version of Figure 5 that demonstrates the cytokine and BDNF increase in the spinal cord of CCI animals.

Minors:

  • Please introduce the abbreviation BDNF.

As you suggested, we have introduced the abbreviation BDNF in the revised version of the manuscript. Please, see page 3 – line 103, in the revised version of the manuscript.

  • Please use consistent units when talking about doses. I would prefer concentrations but I acknowledge that substance amounts are also commonly used. Yet, this paper mixes references of µM , nmol and finally µg when reporting their data and referencing previous studies, which does not allow for clear comparisons.

As you suggest, allowing the reader a comparison between concentrations and other units of measurement, we have reported the molecular weight of the substances used in our experiments. Please, see page 3 – lines 113-114 in the revised version of the manuscript.

  • Please clearly reference shown data in the text with specific labelling of graphs. To this end

figures should have subclassifications (A, B, C… ect)

As you suggest, we have subclassified the new figures as A, B, C, etc. in the revised version of the manuscript.

  • Figure legends: response latency or latency of response would be better terms.

As you suggested, in the figures we used “response latency” in the revision version of the manuscript.

  • It would also be interesting to see if FPR agonists can blunt late phase response in formalin test. Would you check the statistics?

As you suggested, we checked the statistics and we found that the FPR agonists reduced both the early and the late phase of the formalin test. Please, see the result section and the new Figure 2 in the revised version of the manuscript.

  • 7 l.292/293: Which effects were not changed by Boc-1 and WRW4? Does that refer to fMLF treatment?

We are sorry for the mistake. Administration of the Boc-1 and WRW4 antagonists has no effect on carrageenan-induced reduction of the nociceptive threshold. In the new version of the manuscript, we have rewritten the sentence correctly. Please, see page 7 – lines 267-268.

Reviewer 3 Report

In a series of experiments, the authors have reported evidence consistent with a possible role of FPR2/ALX in nociception in the spinal cord. The experiments performed were well described and controlled, and the findings are of interest and add additional information in this area. Statistical significance between controls and experimental data points was achieved in only a few cases. Moreover, in some instances, the statistical differences notwithstanding, the effects were relatively small. Thus, the effects would appear to be more modulatory than regulatory in nature. Such a difference should be emphasized, and to their credit the authors were careful not to over-interpret their results. There are several stylistic issues the authors should consider to improve the manuscript.

  1. The inclusion of long and detailed paragraphs is at times taxing for the reader. It is strongly recommended that the following four paragraphs be broken into one or more paragraphs: lines 61-95, lines 240-272, lines 379-435, and lines 436-489.
  2. The authors should consider a more forceful statement of the purpose of their research. For example, in the last sentence of the Introduction, the wording in line 114, “… further investigated the …” could be rewritten to something like, “… we tested the hypothesis of a …”.
  3. The Results section, in addition to the figures, is replete with detailed factual data throughout, thus making the reading somewhat laborious. The authors should consider the possibility of placing much of the numerical data, including statistical values, in a table(s).

Author Response

We thank the reviewer for helpful comments and suggestions. Our responses to suggestions and comments are shown below in italics. Based on the reviewers' comments and suggestions, we have made significant changes to the text and figures that greatly improved our manuscript.

In a series of experiments, the authors have reported evidence consistent with a possible role of FPR2/ALX in nociception in the spinal cord. The experiments performed were well described and controlled, and the findings are of interest and add additional information in this area. Statistical significance between controls and experimental data points was achieved in only a few cases. Moreover, in some instances, the statistical differences notwithstanding, the effects were relatively small. Thus, the effects would appear to be more modulatory than regulatory in nature. Such a difference should be emphasized, and to their credit the authors were careful not to over-interpret their results. There are several stylistic issues the authors should consider to improve the manuscript.

We thank the reviewer for his kind comments about our work. In any case, we must emphasize that our results indicate that fMLF and BML-111 significantly reduced formalin-induced nociception. BML-111 is also able to induce significant antinociceptive effects in all other pain models used. We think that in order to define the role of FPR2/ALX as modulators or regulators, we would have to conduct other types of experiments. Our first aim was to verify if the FPRs agonists, at the spinal level, could modify nociceptive response and which kind of receptor might be involved. Starting from these results, in a subsequent work, we could conduct experiments aimed at better understanding the modulatory or regulatory role of FPR2/ALX. Regarding the style of the manuscript, we had the new version reviewed by a native English speaker.

The inclusion of long and detailed paragraphs is at times taxing for the reader. It is strongly recommended that the following four paragraphs be broken into one or more paragraphs: lines 61-95, lines 240-272, lines 379-435, and lines 436-489.

As you suggest, in the new version of the manuscript, we have reduced both introduction and discussion and eliminated redundant information (such as the results of statistical analyzes) in the results section.

The authors should consider a more forceful statement of the purpose of their research. For example, in the last sentence of the Introduction, the wording in line 114, “… further investigated the …” could be rewritten to something like, “… we tested the hypothesis of a …”.

As you suggest, we have modified the introduction, trying to make the purpose of our work clearer.

The Results section, in addition to the figures, is replete with detailed factual data throughout, thus making the reading somewhat laborious. The authors should consider the possibility of placing much of the numerical data, including statistical values, in a table(s).

As you suggested, we have deleted data relating to statistical analyzes in the results section to make it easier to read. We do not consider it useful to present the deleted statistical data in a table, as the statistical analyzes carried out are reported in the captions of the figures and the results obtained are shown graphically in the figures.

Round 2

Reviewer 1 Report

The authors explored the role of formyl peptide receptor type 2 (FRP2) in nociception. Presented fairly comprehensive data suggest that FRP2 activation in the spinal cord suppresses pain response, at least in some experimental models. The findings are novel and might have therapeutic implications.

The manuscript was greatly improved in revision, all my concerns were satisfactorily addressed.

Reviewer 2 Report

All my points have been adequately addressed.